# Robust Learning against Relational Adversaries

**Yizhen Wang**
Visa Research
yizhewan@visa.com

**Mohannad Alhanahnah**
University of Wisconsin–Madison
alhanahnah@wisc.edu

**Xiaozhu Meng** *
Rice University
Xiaozhu.Meng@rice.edu

**Ke Wang**
Visa Research
kewang@visa.com

**Mihai Christodorescu** †
Visa Research
mihai.christodorescu@visa.com

**Somesh Jha**
University of Wisconsin–Madison
jha@cs.wisc.edu

## Abstract

Test-time adversarial attacks have posed serious challenges to the robustness of machine-learning models, and in many settings the adversarial perturbation needs not be bounded by small $\ell_p$-norms. Motivated by attacks in program analysis and security tasks, we investigate *relational adversaries*, a broad class of attackers who create adversarial examples in a reflexive-transitive closure of a logical relation. We analyze the conditions for robustness against relational adversaries and investigate different levels of robustness-accuracy trade-off due to various patterns in a relation. Inspired by the insights, we propose *normalize-and-predict*, a learning framework that leverages input normalization to achieve provable robustness. The framework solves the pain points of adversarial training against relational adversaries and can be combined with adversarial training for the benefits of both approaches. Guided by our theoretical findings, we apply our framework to source code authorship attribution and malware detection. Results of both tasks show our learning framework significantly improves the robustness of models against relational adversaries. In the process, it outperforms adversarial training, the most noteworthy defense mechanism, by a wide margin.

## 1 Introduction

The robustness of machine learning (ML) systems has been challenged by test-time attacks using adversarial examples [Szegedy et al., 2013]. These adversarial examples are intentionally manipulated inputs that preserve the essential characteristics of the original inputs, and thus are expected to have the same test outcome as the originals by human standard; yet they severely affect the performance of many ML models across different domains [Moosavi-Dezfooli et al., 2016, Eykholt et al., 2018, Qin et al., 2019]. As models in high-stake domains such as system security are also undermined by attacks [Grosse et al., 2017, Rosenberg et al., 2018, Hu and Tan, 2018, Pierazzi et al., 2020], robust ML in adversarial test environment becomes an imperative task for the ML community.

Existing work on test-time attack and defense evaluation predominately considers $\ell_p$-norm bounded adversarial manipulation [Goodfellow et al., 2014, Carlini and Wagner, 2017, Papernot et al., 2018,

---

*This work was done when the author was at Rice University.

†This work was done when the author was at Visa Research. The author is now at Google.

36th Conference on Neural Information Processing Systems (NeurIPS 2022).

Nicolae et al., 2018]. However, in many security-critical settings, the adversarial examples need not respect the $\ell_p$-norm constraint as long as they preserve the malicious behavior. For example, in malware detection, a malware author can implement the same function using different APIs, or bind a malware within benign software like video games or office tools. The modified malware preserves the malicious functionality despite the drastically different syntactic features. Hence, focusing on adversarial examples of small $\ell_p$-norm in this setting will fail to address a sizable attack surface that attackers can exploit to evade detectors.

In this paper, we consider a general threat model — the relational adversary — in which the attacker can manipulate the original test inputs via transformations specified by a logical relation. Unlike the prior work [Rosenberg et al., 2018, Hu and Tan, 2018, Hosseini et al., 2017, Hosseini and Poovendran, 2018] which investigates specific adversarial settings, our paper extends the scope of attacks to general logical transformation, which can be readily instantiated to incorporate real-world transformations over different domains. Moreover, the relational adversary can apply an arbitrary sequence of transformations to the inputs as long as the essential semantics of the input is preserved.

From the defense perspective, recent work has started to look beyond $\ell_p$-norm constraints, including adversarial training [Grosse et al., 2017, Rosenberg et al., 2019, Lei et al., 2019, Laidlaw et al., 2020, Li et al., 2022], verification-loss regularization [Huang et al., 2019] and invariance-induced regularization [Yang et al., 2019]. Adversarial training in principle can achieve high robust accuracy when the adversarial example in the training loop maximizes the loss. However, finding such adversarial examples is in general NP-hard [Katz et al., 2017] and even PSPACE-hard for attacks considered in this paper (Appendix A.1). Huang et al. [2019] and Yang et al. [2019] add regularizers which incorporate model robustness as part of the training objective. However, such regularization do not strictly enforce the robustness, as a result, models are still vulnerable.

***Normalize-and-Predict Learning Framework*** This paper attempts to overcome the limitations of prior work by introducing a learning framework, *normalize-and-predict* (hereinafter abbreviated as *N&P*), which guarantees robustness by design. Unlike the prior work which exclusively consider the $\ell_p$-norm bounded attacks, we target a *relational* adversary, whose admissible manipulation is specified by a logical relation. We consider a strong adversary who can apply an arbitrary number of transformations. The key idea underlying *N&P* is *normalization*, a powerful concept in computer science — canonicalizing data with multiple representations into a unique form — that is widely applied across various domains (e.g. canonicalization of filenames for computer security, lemmatisation in computational liguistics, etc.). Technically, *N&P* first converts each data point to a canonical form and subsequently restricts the training and test of models on the normalized data. A potential downside of *N&P* is the sacrifice of model accuracy in exchange of guaranteed robustness. To cope with this issue, we propose a unified framework that combines *N&P* with adversarial training in an attempt to achieve the optimal robust-accuracy trade-off. Specifically, the unified framework selectively normalizes relations over which model accuracy can be preserved and adversarially trains on the rest. Our unified framework gets the benefits from both *N&P* and adversarial training.

We evaluate *N&P* in the settings of source code authorship attribution and malware detection. For the former, we set out to defend two attribution approaches Caliskan-Islam et al. [2015], Abuhamad et al. [2018] against the state-of-the-art attack proposed by Quiring et al. [2019]. For the latter, we first formulate two types of common program transformation — (1) addition of redundant libraries and API calls, and (2) substitution of equivalent API calls — as logical relations. Next, we propose two generic relational adversarial attacks to determine the robustness of a model.

The results we obtained in both tasks show that:

1. Models obtained by *N&P* and the unified framework achieve significantly higher robust accuracy than the vanilla models. In particular, the improvement of robustness far outweighs the drop in accuracy on clean inputs, suggesting a worthwhile trade-off when a sizable portion of the input comes from adversarial sources.

2. *N&P* achieves higher robust accuracy than adversarial training especially when attackers use a stronger or different attack method than the one used in *N&P* or adversarial training.

3. Compared to adversarial training, *N&P* incurs a substantially lower computation overhead when defending against *problem space* attacks, where adversarial examples are generated through program transformation on the raw code samples rather than gradient manipulation on input features.

Finally, based on our theoretical and empirical results, we conclude that input normalization is vital to robust learning against relational adversaries. We believe techniques that can improve the quality of normalization are promising directions for future work.

## 2  Related Work.

Test-time attacks using adversarial examples have been extensively studied in the past several years. Research has shown ML models are vulnerable to such attack in a variety of application domains [Moosavi-Dezfooli et al., 2016, Chen et al., 2017, Papernot et al., 2017, Eykholt et al., 2018, Ebrahimi et al., 2018, Qin et al., 2019, Yang et al., 2020a] including system security and program analysis where reliable defense is essential. For instance, Grosse et al. [2017] and Al-Dujaili et al. [2018] evade API/library usage based malware detectors by adding redundant API calls; Rosenberg et al. [2018], Hu and Tan [2018], and Rosenberg et al. [2019] successfully attack running-time behavior based detectors by adding redundant execution traces; Quiring et al. [2019] shows semantic-preserving edits over source code can evade authorship attribution. Pierazzi et al. [2020] extend the attacks from feature-space to problem-space to create realistic, executable attack instances using automated software transplantation.

On the defense end, adversarial training has been the most widely used approach [Grosse et al., 2017, Al-Dujaili et al., 2018, Rosenberg et al., 2019, Li et al., 2022]. In particular, Al-Dujaili et al. [2018] and Li et al. [2022] attempt adversarial training on the same malware detection and authorship attribution tasks in our experiment evaluation, respectively. We show that adversarial training is hard to optimize and ineffective against different attack algorithms or attacks with stronger search parameters than in training. Yang et al. [2019] adds invariance-induced regularizers which indicate the worst-case loss of searching over input transformations. Hendrycks et al. [2019] proposes self-supervised learning in which the learner tries to identify the transformations applied to the training inputs. We extend the scope from specific spatial transformation attacks in image classification to a general adversary based on logic relations. We note that regularization and self-supervised learning may not enforce model robustness on finite samples, nor will they be computationally efficient over arbitrarily long sequence of input transformations. In contrast, *N&P* enforces robustness by design. Incer et al. [2018], Kouzemtchenko [2018] enforce monotonicity over model outputs so that the addition of feature values always increase the maliciousness score. These approaches are limited to guarding against the addition attacks, thus lacks generality.

Normalization is a technique to reduce the number of syntactically distinct instances. First introduced to network security in the early 2000s in the context of intrusion detection systems [Handley et al., 2001], it was later applied to malware detection [Christodorescu et al., 2007, Coogan et al., 2011, Bichsel et al., 2016, Salem and Banescu, 2016, Baumann et al., 2017]. Our work addresses the open question whether normalization is useful for ML under relational adversary by investigating its impact on both model robustness and accuracy.

The robustness-accuracy trade-off against $\ell_p$-attacks has been discussed in literature [Zhang et al., 2019, Tsipras et al., 2018, Fawzi et al., 2018, Raghunathan et al., 2020, Yang et al., 2020c]. The trade-off can either be 1) distributional, where inputs with different Bayes-optimal labels are mixed up in the adversarial feasible set, or 2) algorithmic, where the trade-off is due to low model expressiveness, small sample size, inductive bias in training data or deficiency in the learning algorithm. The former is inevitable because no classifier can achieve both optimal robustness and accuracy. Zhang et al. [2019] construct such an example distribution: the real number line is divided into intervals of size $\epsilon$, and neighboring segments always have different labels. Our analysis focuses on distributional trade-off because *N&P* guarantees robustness against the normalized transformations, i.e. there is no further algorithmic trade-off if normalization is exact. The analysis helps strategically choose the set of transformations to normalize in the unified framework. We also note that *N&P* can use any existing training/testing procedure except the inputs are now normalized. Therefore, existing $\ell_p$-based trade-off analysis also applies on the normalized input space for $\ell_p$-attacks.

## 3  Threat Model

We consider a data distribution $\mathcal{D}$ over an input space $\mathcal{X}$ and categorical label space $\mathcal{Y}$. We use bold face letters, e.g. $\mathbf{x}$, for input vectors and $y$ for the label. Given a hypothesis class $\mathcal{H}$, the learner

```
1 for (i = k; i > 0; i--)         p₀
2    printf("%d", ans[i]);
3 bool a = 0;
```

```
1 for (i = k; i > 0; i--)         p₁
2    cout << ans[i];
3 bool a = 0;
```

```
1 i = k;                          p₂
2 while (i > 0) {
3    cout << ans[i];
4    i--;
5 }
6 bool a = 0;
```

```
1 i = k;                          p₃
2 while (i > 0) {
3    cout << ans[i];
4    i--;
5 }
6 bool a = false;
```

Figure. 1: An example of semantic-preserving code transformations. Each arrow indicates one transformation, and all code pieces are semantically equivalent.

wants to learn a classifier $f : \mathcal{X} \to \mathcal{Y}$ in $\mathcal{H}$ that minimizes the risk over the data distribution. In non-adversarial settings, the learner solves $\min_{f \in \mathcal{H}} \mathbb{E}_{(\mathbf{x},y) \sim \mathcal{D}} \ell(f, \mathbf{x}, y)$, where $\ell$ is a loss function. For classification, $\ell(f, \mathbf{x}, y) = \mathbb{1}(f(\mathbf{x}) \neq y)$.

***Logical Relation.*** A relation $\mathcal{R}$ is a set of input pairs. Each pair $(\mathbf{x}, \mathbf{z}) \in \mathcal{R}$ implies a viable transformation from $\mathbf{x}$ to $\mathbf{z}$. We write $\mathbf{x} \to_{\mathcal{R}} \mathbf{z}$ iff $(\mathbf{x}, \mathbf{z}) \in \mathcal{R}$, and write $\mathbf{x} \to_{\mathcal{R}}^* \mathbf{z}$ iff $\mathbf{x}$ can arrive at $\mathbf{z}$ via an arbitrary number of transformations specified by $\mathcal{R}$. In other words, $\to_{\mathcal{R}}^*$ is the reflexive-transitive closure of $\to_{\mathcal{R}}$. We describe an example relation as follows:

**Example 1** (Semantic-preserving Program Transformations). *Let $P$ denote a set of programs and $T$ be a set of transformations that preserves program semantics. Then $T$ induces a relation $\mathcal{R} = \{(p, t(p)) | \forall p \in P, t \in T\}$ on the space $P$. Figure 1 shows a concrete example. The initial program $p_0$ undergoes (1)* for-to-while *(2)* printf-to-cout, *and (3)* 0/1-to-false/true *transformations to become $p_3$. We have $p_i \to_{\mathcal{R}} p_{i+1}$ and $p_i \to_{\mathcal{R}}^* p_j$ for $i \in \{0, 1, 2\}, j \geq i$.*

***Attacker's Capability.*** A test-time adversary replaces a clean test input $\mathbf{x}$ with an adversarially manipulated input $A(\mathbf{x})$, where $A(\cdot)$ represents the attack algorithm that searches for adversarial examples in a feasible set $\mathcal{T}(\mathbf{x})$. We consider an adversary who wants to maximize the classification error rate: $\mathbb{E}_{(\mathbf{x},y) \sim \mathcal{D}} \mathbb{1}(f(A(\mathbf{x})) \neq y)$. We assume *white-box* attacks[3], i.e. the adversary has total access to $f$, including its structures, model parameters and any defense mechanism in place.

**Definition 1** (relational adversary). *An adversary is $\mathcal{R}$-relational if $\mathcal{T}(\mathbf{x}) = \{\mathbf{z} \mid \mathbf{x} \to_{\mathcal{R}}^* \mathbf{z}\}$.*

In essence, a relational adversary can apply arbitrary composition of transformations specified in $\mathcal{R}$. We assume $\mathcal{R}$ to be accessible to both the learner and the adversary. We believe this is a reasonable assumption to make: $\mathcal{R}$ determines the adversary's search space like $\epsilon$ and $p$ for $\ell_p$-norm attacks, and it is impossible to formally evaluate a defense technique against unknown adversarial feasible sets.

***Challenges with Relational Adversaries.*** Relational adversaries are prevalent in program analysis because 1) the attacker often has enough control over the victim input to make substantial changes, and 2) well-defined semantic-preserving transformations exist. We will focus on two such threats — misleading authorship attribution [Quiring et al., 2019] and evading malware detection [Al-Dujaili et al., 2018] — in the empirical evaluation in Sec 5. In the former, the attacker transforms source code samples so as to evade automated authorship attribution while maintains the behavior of the code. In the latter, a malware author manipulates the API usage to evade ML-based detectors while maintains the malicious functionality.

Relational adversaries pose new challenges to robust learning. First, since $\mathcal{T}(\mathbf{x})$ is discrete in nature and the perturbation need not be bounded by $\ell_p$-norm, defense mechanisms that leverage local smoothness of model prediction are no longer applicable. Second, adversarial training specifically suffers from efficiency issues against relational adversaries. The reasons are two two-fold. First, the inner maximization procedure needs to search a combinatorially large discrete space. Second, the generation of adversarial examples is based on program transformation in the problem space[4] (rather than gradient manipulation in the feature space), which can not be performed in GPU. These challenges motivate us to study the essence of $\mathcal{R}$ and leverage normalization as a remedy (Sec 4). We evaluate models by the robustness and robust accuracy adapted for relational adversaries as follows,

---

[3]We consider a strong white-box attacker to avoid interference from security by obscurity, which is shown fragile in various other adversarial settings [Carlini and Wagner, 2017].

[4]This is to ensure the syntactic validity of the generated program.

**Definition 2** (Robustness and robust accuracy). *Let $Q(\mathcal{R}, f, \mathbf{x})$ be the following statement: $\forall \mathbf{z}((\mathbf{x} \to_{\mathcal{R}}^* \mathbf{z}) \Rightarrow f(\mathbf{x}) = f(\mathbf{z}))$. Then, a classifier $f$ is robust at $\mathbf{x}$ if $Q(\mathcal{R}, f, \mathbf{x})$ is true, and the robustness of $f$ to an $\mathcal{R}$-relational adversary is: $\mathbb{E}_{\mathbf{x} \sim \mathcal{D}_{\mathcal{X}}} \mathbb{1}_{Q(\mathcal{R}, f, \mathbf{x})}$, where $\mathbb{1}_{(\cdot)}$ indicates the truth value of a statement and $\mathcal{D}_{\mathcal{X}}$ is the marginal distribution over inputs. The robust accuracy of $f$ w.r.t. an $\mathcal{R}$-relational adversary is then: $\mathbb{E}_{(\mathbf{x}, y) \sim \mathcal{D}} \mathbb{1}_{Q(\mathcal{R}, f, \mathbf{x}) \wedge f(\mathbf{x}) = y}$.*

## 4  *N&P*– A Robust Learning Framework

The *Normalize-and-Predict* (*N&P*) framework enhances model robustness by learning and testing over normalized training and test inputs. The framework originates from the following principle:

> *Suppose we can convert a test input into some canonical form — the **normal** form — and use the normal form as the model input, then the model prediction is robust if the adversarial example and the original clean input share the same normal form.*

We refer the conversion of an input to its normal form as ***normalization***. In this section, we answer the crucial questions of *when* and *how* to normalize. We first introduce the framework in Sec 4.1 and then theoretically analyze its performance in light of robustness-accuracy trade-off in Sec 4.2. The analysis shows that a carefully chosen normal form can help achieve the optimal robust accuracy. Following these insights, we show a unified framework of *N&P* with adversarial training in Sec 4.3 and a computationally efficient heuristic normalizer in Sec 4.4.

### 4.1  An Overview of the *N&P* Framework

In *N&P*, the learner first specifies a normalizer $\mathcal{N} : \mathcal{X} \to \mathcal{X}$. We call $\mathcal{N}(\mathbf{x})$ the 'normal form' of input $\mathbf{x}$. The learner then both trains the classifier and predicts the test label over the normal forms instead of the original inputs. Let $D$ denote the training set. In the empirical risk minimization learning scheme, for example, the learner will now solve the following problem

$$\min_{f \in \mathcal{H}} \sum_{(\mathbf{x}, y) \in D} \ell(f, \mathcal{N}(\mathbf{x}), y), \tag{1}$$

and use the minimizer $f^*$ as the classifier. For an actual learning algorithm, the *N&P* pipeline will replace the training input $(\mathbf{x}, y)$ with $(\mathcal{N}(\mathbf{x}), y)$. At test-time, the model will predict $f^*(\mathcal{N}(\mathbf{x}))$, i.e. replace the original test input $\mathbf{x}$ with the normal form $\mathcal{N}(\mathbf{x})$.

### 4.2  Finding the Normalizer — Trade-off Analysis

The choice of $\mathcal{N}$ is crucial to *N&P*'s performance in terms of the *robustness-accuracy trade-off*. For an input $\mathbf{x}$, let $S_{\mathcal{N}}(\mathbf{x}) = \{\mathbf{z} \in \mathcal{X} \,|\, \mathcal{N}(\mathbf{x}) = \mathcal{N}(\mathbf{z})\}$ denote the set of inputs that share the same normal form as $\mathbf{x}$. For robustness purpose, we want $S_{\mathcal{N}}(\mathbf{x})$ to be inclusive: if $S_{\mathcal{N}}(\mathbf{x})$ covers the adversary's feasible set $\mathcal{T}(\mathbf{x})$, then the model will be robust at $\mathbf{x}$ by design. Meanwhile, for accuracy purpose, we want to restrict the size of $S_{\mathcal{N}}(\mathbf{x})$: a constant $\mathcal{N}$ is robust, but has no utility as $f(\mathcal{N}(\cdot))$ is also constant. Therefore, we seek an $\mathcal{N}$ that performs only the necessary normalization for robustness with a minimal impact on accuracy.

***Price for Robustness.*** Like against $\ell_p$ attacks, robustness-accuracy trade-off also exists for learning against relational attacks. We first examine the intrinsic trade-off due to the structure of the relation. Given a relation $\mathcal{R}$, we have the following condition for a classifier to be robust everywhere.

**Proposition 1.** *A classifier $f$ is robust at all $\mathbf{x} \in \mathcal{X}$ iff $\mathbf{x} \to_{\mathcal{R}} \mathbf{z} \implies f(\mathbf{x}) = f(\mathbf{z})$ for all $\mathbf{x} \in \mathcal{X}$.*

Under this condition, we characterize three interesting patterns of $\mathcal{R}$ that cause different levels of trade-off as shown in Figure 2. First, if $\mathbf{x} \to_{\mathcal{R}} \mathbf{z}$ and $\mathbf{x}, \mathbf{z}$ have the same label, then a robust classifier $f$ comes at no cost of natural accuracy. Second, if $\mathbf{x}, \mathbf{z}$ have different labels but can be transformed into each other under $\mathcal{R}$, then a robust $f$ will have to sacrifice the natural accuracy for either $\mathbf{x}$ or $\mathbf{z}$. Fortunately, as we will explain in Theorem 1, such cost to natural accuracy is indeed necessary for achieving the best *robust* accuracy. Last, if $\mathbf{x}$ can be transformed to two outputs $\mathbf{z}_1, \mathbf{z}_2$ with different labels, then enforcing robustness may cause additional trade-off to robust accuracy.

We note that all three patterns are common in program analysis tasks. The second pattern happens when some syntactical feature is strongly correlated with the class label. For example, a malware can

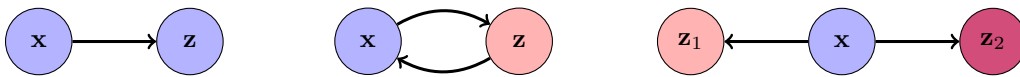

Figure. 2: Relations with different robustness-accuracy trade-off. Different node colors indicate different most likely labels. **Left:** Robust classification preserves natural accuracy; **Middle:** Robust classification preserves robust accuracy; **Right:** Robust classification is at odds with robust accuracy.

call two different APIs — one is open-sourced and the other is a secured version with authentication — for the same function. Malware authors at large predominately use the former for convenience, and thus make its usage a strong signal of malware. However, an advanced attacker may compromise the authentication and subsequently uses the secured API to evade detection. *N&P* avoids using such features and thus corrects the false sense of security. The third pattern often happens with code injection: $\mathbf{x}$ could be an abstract function, while $\mathbf{z}_1, \mathbf{z}_2$ are two instantiations with different purposes. Enforcing the same prediction over $\mathbf{x}, \mathbf{z}_1$ and $\mathbf{z}_2$ makes no sense in most cases.

***Optimal $\mathcal{N}$ for Robust Accuracy.*** Inspired by the robustness-accuracy trade-off analysis, we discover the following normalizer $\mathcal{N}$ that preserves the *optimal* robust accuracy before and after normalization.

**Definition 3** (Equivalence Group). *A set $\mathcal{E} \subseteq \mathcal{X}$ is an equivalence group under relation $\mathcal{R}$ iff 1) $\forall \mathbf{x}, \mathbf{z} \in \mathcal{E}$, we always have $\mathbf{x} \to_{\mathcal{R}}^* \mathbf{z}$ and $\mathbf{z} \to_{\mathcal{R}}^* \mathbf{x}$, and 2) $\forall \mathbf{x}, \mathbf{z} \in \mathcal{X}, (\mathbf{x} \to_{\mathcal{R}}^* \mathbf{z}) \wedge (\mathbf{z} \to_{\mathcal{R}}^* \mathbf{x})$ implies $\mathbf{x}, \mathbf{z} \in \mathcal{E}$. In addition, a single-element set $\{\mathbf{x}\}$ is an equivalence group if $\mathbf{x}$ does not belong to any equivalence group with more than one element.*

**Definition 4** (Normalizer by Equivalence Group). *A normalizer by equivalence group $\mathcal{N}$ picks a deterministic element $\mathbf{z}$ for each equivalence group $\mathcal{E}$, and returns $\mathcal{N}(\mathbf{x}) = \mathbf{z}$ for all $\mathbf{x} \in \mathcal{E}$.*

**Theorem 1** (Preservation of Robust Accuracy). *Let $\mathcal{H}$ be the set of all labeling functions and $\mathcal{N}$ be a normalizer by equivalence group, then we must have a classifier $f \in \mathcal{H}$ such that 1) $f$ has the highest robust accuracy on $\mathcal{D}$ without normalization, 2) $f(\mathbf{x}) = f(\mathbf{z})$ for all $\mathbf{x}, \mathbf{z}$ in the same equivalence group, and 3) there exists a classifier $g \in \mathcal{H}$ over the normalized inputs such that $g(\mathcal{N}(\mathbf{x})) = f(\mathbf{x}), \forall \mathbf{x} \in \mathcal{X}$.*

Theorem 1 convey an important message. The robustness guaranteed by *N&P* with a normalizer in Definition 4, i.e. same prediction for all $\mathbf{x}$ in the same equivalence group, is also desired by some model that achieve the highest robustness accuracy on the original unnormalized inputs. Therefore, learning and testing on the normalized inputs incurs no additional robust-accuracy trade-off.

### 4.3 Synergy with Adversarial Training

Despite the difference between the underlying principles of *N&P* and adversarial training, we can readily unify these two approaches to enjoy the benefits from both worlds. Let $A(\cdot)$ denote the attack algorithm used in the inner loop of adversarial training. In the unified framework, the learner solves the min-max loss of adversarial training over the *normalized* inputs, i.e.

$$\min_{f \in \mathcal{H}} \max_{A(\cdot)} \sum_{(\mathbf{x}, y) \in D} \ell\left(f, A\left(\mathcal{N}(\mathbf{x})\right), y\right) \tag{2}$$

to obtain a model $f^*$ and predicts $f^*(\mathcal{N}(\mathbf{x}))$ at test-time. In the actual learning algorithm, the learner

- first normalizes all training examples with an $\mathcal{N}$ by equivalence groups, and
- in each iteration, trains over the adversarial variant of the *normalized* input $(\mathcal{N}(\mathbf{x}), y)$.

Normalization offers many advantages to the unified approach. First, the unified framework still guarantees robustness for adversarial examples in the same equivalence group as the clean input. Second, normalization significantly reduced the search space of adversarial training. Suppose a program uses $n$ APIs and each API has $k$ functionally equivalent substitutes, then the number of variants by API substitution alone grows exponentially to $k^n$. Normalization removes these variants so that adversarial training can focus on other transformations. Third, normalization can potentially reduce the model capacity needed to learn a robust model as shown in Appendix A.2. In return, adversarial training can deal with transformations for which normalization incurs additional robustness-accuracy trade-off, e.g. against code injection as shown in Sec 4.2.

### 4.4 Efficient Normalization

Although the problem of exact normalization in its most general form can be computationally hard, we propose a practical framework of generating heuristic normal forms. First, for each equivalence group $\mathcal{E}$, we assign an *order* to all its elements. An ideal normalizer should return the element with the *lowest* order. Given an input, our heuristic normalizer will iteratively apply the transformation that *lowers* the order of the input until no such transformations are available. The final form will be the approximate normal form. Taking semantically equivalent source codes in authorship attribution as an example. We can rank code pieces by its syntactical features, e.g. number of for loops. The normalizer, while trying to reduce the order, will apply the *for-to-while* transformation. To ensure that the approximate normal form still lies in the same equivalence group, every transformation we use in $\mathcal{N}$ has a reversible counterpart in $\mathcal{R}$, e.g. *for-to-while* can be negated by *while-to-for*. Then, $\mathcal{N}(\mathbf{x})$ can always be converted back to $\mathbf{x}$ by applying the transformations in the reverse direction. The detailed description can be found in Appendix B. The code of our normalizer is also open sourced on github. [5]

## 5 Experiment

We now evaluate the effectiveness of *N&P* against relational attacks for real-world attacks. In particular, we seek answers to the following questions.

1. Does *N&P* deliver the promised robustness improvement under real-world attacks?

2. How much performance edge does *N&P* provide — both in robust accuracy and computation time — compared to standard adversarial training against relational adversaries?

3. How is the robustness-accuracy trade-off compared to without normalization, and is the trade-off worthwhile?

We investigate these questions over two real-world tasks — authorship attribution and malware detection. Our result shows that normalization in the learning pipeline can significantly boost model robustness against relational adversaries compared to adversarially trained and unprotected models.

### 5.1 Authorship Attribution

Automated authorship attribution is a classical task in program analysis. Successful identification of the author of a code piece can help catching plagiarism, tracking contributors to shared projects and identify authors of malicious content [Caliskan-Islam et al., 2015, Abuhamad et al., 2018]. However, Quiring et al. [2019] shows that semantic-preserving transformation over the source code can significantly reduce the performance of ML-based authorship attribution models. Although the attack is hard to defend if the attacker knows the coding style of the imitation target well, we are interested in whether authorship attribution is still possible under common code transformations that does *not* require any knowledge of the imitation target.

***Dataset.*** We use the dataset provided by Quiring et al. [2019], [6] which is collected from Google Code Jam, a coding platform on which individual programmers compete to solve coding challenges. It consists of 1,632 files of C++ code from 204 authors solving the same 8 programming challenge questions. We follow the same train-test data splits in Quiring et al. [2019]. We create 8 different data splits. Each split uses the codes from one challenge as the test set and the codes from the rest seven challenges for training. We run the experiments over all 8 splits and report the average results.

***Relation and Normalization.*** The attack proposed by Quiring et al. [2019] applies Monte-Carlo tree search to determine the sequence of program transformations for creating adversarial examples. As explained in Quiring et al. [2019], all transformations, 42 in total covering control-flow, variable and function declaration, I/O API, etc., merely change the common, generic program features that do not fundamentally alter the signature of each author, therefore, the generated forgeries should have the same label as the original copies. To keep our engineering workload manageable, we consider 35 transformations that are easier to normalize. Some of the transformations are shown in

---

[5]https://github.com/Mohannadcse/Normalizer-authorship
[6]https://github.com/EQuiw/code-imitator/tree/master/data/dataset_2017

Table 1: Authorship attribution accuracy(%). The leader for each series is highlighted in bold.

| Attack methods | N&P-RF | N&P-LSTM | Adv-LSTM | Vanilla-RF [Caliskan-Islam et al., 2015] | Vanilla-LSTM [Abuhamad et al., 2018] |
|---|---|---|---|---|---|
| Clean | 76.1±3.8 | 78.7±4.8 | 76.7±3.8 | **90.4**±1.7 | 88.4±3.7 |
| Non-Adaptive | 72.3±4.0 | **73.7**±3.6 | 30.8±4.0 | 13.2±3.3 | 21.1±2.8 |
| Adaptive | 70.5±4.6 | **71.2**±4.8 | 30.8±4.0 | 13.2±3.3 | 21.1±2.8 |
| Adaptive+ | 37.3±13.3 | **49.9**±7.4 | 25.2±5.1 | 1.0±0.4 | 0.9±0.4 |

Figure 1. Regarding the normalization procedure, we use the sum of selected syntactical features of the code (e.g. number of *for* loops, C-style I/O API and Boolean variables) as the order of the code. We iteratively apply transformations that lowers the order until reaching a fixed point. The full transformation and syntactical features list are in Appendix B.

***Attack Methods.*** We consider three attack modes. All three attacks use the Monte-Carlo tree search (MCTS) method in Quiring et al. [2019] with the same parameters including number of iterations and roll-out per iteration. The **Non-Adaptive** method attacks without knowing the existence the normalizer $\mathcal{N}$, i.e. runs MCTS using prediction scores over the original inputs. The **Adaptive** attack, in contrast, runs MCTS using the model prediction over the normalized inputs. This means for each roll-out attempt, the attacker will also run the normalizer and using the prediction score over the normalized input to plan its next move. Last, we consider **Adaptive+**, which runs MCTS over normalized inputs and also attacks with *all* 42 transformations in Quiring et al. [2019]. Although no performance is guaranteed against an attacker with a larger adversarial feasible set than in training, we are still curious about how *N&P* compares to the adversarial trained and the unprotected models.

***Baseline Models.*** We consider both the random forest (RF) model and the recurrent neural net model with LSTM units attacked in Quiring et al. [2019] for baseline performance. Hereinafter, we name them **Vanilla-RF** and **Vanilla-LSTM**. Our *N&P* framework uses the same model structure, parameters and training procedure as Quiring et al. Hereinafter, we call the normalized models *N&P*-**RF** and *N&P*-**LSTM**. For the adversarial training baseline, we train the lone compatible model **Vanilla-LSTM** into **Adv-LSTM** as **Vanilla-RF**, a non-parametric method, has no gradients for adversarially training to exploit [Wang et al., 2018, Yang et al., 2020b]. We note that the standard adversarial training is too computationally expensive for the attack on source code level. We make a number of adaptations that reduce the number of MCTS roll-outs and generate adversarial examples in batch for better parallelism so that the process finishes within a month on a 72-core CPU server. The details of these adaptations can be found in Appendix B.

***N&P v.s. Vanilla Models.*** Table 1 shows the test accuracy of all baselines against adversarial examples and clean inputs. *N&P*-**LSTM** has the highest accuracy against all three attacks followed by *N&P*-**RF**. Compared to the corresponding vanilla models with the same model structures, the *N&P* models achieves higher accuracy by a wide margin. The accuracy increases by more than 50% for both the non-adaptive and adaptive attack. The results show that *N&P* is highly effective when the attacker uses transformations already considered by the normalizer. Intriguingly, even under the Adaptive+ attack in which some attack transformations are not normalization, *N&P* still achieves nontrivial accuracy – 37.3% for *N&P*-**RF** and 49.9% for *N&P*-**LSTM** – which is 36% and 49% higher than **Vanilla-RF/Vanilla-LSTM**. This is because most of the time the MCTS attack will use the 7 transformations that have not been normalized. *N&P* effectively reduces the attack surface and thus still enhances the accuracy. For clean inputs, the *N&P* models has lower accuracy compared to the vanilla models due to the inevitable robustness-accuracy trade-off analyzed in Sec 4.2. However, the difference (<10% for LSTM and <15% for RF) is much smaller than the accuracy gain in the adversarial setting. The trade-off is worthwhile when a sizable portion of the inputs come from adversarial sources.

***N&P v.s. Adversarial Training.*** *N&P* models also consistently outperforms the adversarially trained counterparts across all attacks by a significant margin (~40% for Non-Adaptive/Adaptive and up to 24% for Adaptive+). The performance of adversarial training is heavily affected by the strength of the attack used in the training loop: a model adversarially trained with a weak attack in the loop may succumb to a strong attack in test time. Recall that our adversarial training procedure uses an attack with reduced search parameters in order to have reasonable training time. **Adv-LSTM** has ~60% accuracy against the adaptive attack in the training loop. However, the accuracy drops significantly to

Table 2: Malware Detection: False Negative Rate (FNR) and False Positive Rate (FPR) on *Sleipnir*.

| | Unified (Ours) | | Adv-Trained | | Al-Dujaili et al. [2018] | | Natural | |
|---|---|---|---|---|---|---|---|---|
| | FNR(%) | FPR(%) | FNR(%) | FPR(%) | FNR(%) | FPR(%) | FNR(%) | FPR(%) |
| Natural | 5.0±0.4 | 11.9±1.2 | 5.8±0.9 | 12.1±1.2 | 6.4±0.5 | 10.7±0.3 | 6.2±0.6 | 10.0±0.6 |
| Adversarial | 5.5±0.5 | 11.9±1.2 | 27.9±8.2 | 12.1±1.2 | 89.9±7.8 | 10.7±0.3 | 100±0.0 | 10.0±0.6 |

30.8% against the full-strength MCTS attack in the actual test. In security applications, the attacker is often assumed to have more computation power than the defender and may use attack algorithms unknown to the defender in which case *N&P* is likely to show more consistent performance.

***Running Time.*** The vanilla models take less than 12 hours to train on our infrastructure. *N&P* incurs an overhead of less than 12 hours in normalization, which is in the same order of the vanilla training time. In contrast, adversarial training requires much longer training time. Even with reduced search parameters, adversarial training still takes more than 20 days to finish on the same infrastructure, which is 40x more than the vanilla training time. *N&P* shows a clear advantage in running time.

## 5.2 Malware Detection

***Dataset.*** We evaluate the effect of normalization on malware detection using *Sleipnir*, a data set containing Windows binary API usage features of 34,995 malware and 19,696 benign software, extracted from their Portable Executable (PE) files using LIEF [Thomas, 2017]. The dataset was created by Al-Dujaili et al. [2018] and used to evaluate the effectiveness of their adversarial training against API injection attacks. The detection is exclusively based on the API usage of a malware. There are 22,761 unique API calls in the data set, so each PE file is represented by a binary indicator vector $\mathbf{x} \in \{0,1\}^m$, where $m = 22,761$. We sample 19,000 benign PEs and 19,000 malicious PEs to construct the training (60%), validation (20%), and test (20%) sets.

***Relation and Normalization.*** Al-Dujaili et al. [2018] considers adding redundant API calls, i.e., $(\mathbf{x}, \mathbf{z}) \in \mathcal{R}$ *iff* $\mathbf{z}$ is obtained by flipping some $\mathbf{x}$'s feature values from 0 to 1. On top of API addition, We consider substitution of API calls with functionally equivalent counterparts, i.e., $(\mathbf{x}, \mathbf{z}) \in \mathcal{R}$ *iff* $\mathbf{z}$ is obtained by changing some of $\mathbf{x}$'s feature values from 1 to 0 in conjunction with some other feature values changed from 0 to 1. With expert knowledge, we extract nearly 2,000 equivalent API groups described in Appendix C.3. We normalizes API substitutions by condensing the features of equivalent APIs into one whose value indicates if *any* API in the group is used.

***Attack Methods.*** We introduce two new relational attack algorithms, which are GREEDYBYGROUP and GREEDYBYGRAD. GREEDYBYGROUP searches the combination of API usage within each equivalence group that maximizes the test loss, and then combine the adversarial perturbations from all equivalence groups. GREEDYBYGRAD makes a first-order approximation of the change in test loss caused by potential transformations, applies the transformations with top $m$ approximated increases and repeats this procedure for $K$ iterations. Their detailed algorithm descriptions are in Appendix C.1. In model evaluation, we use our attacks together with the `rfgsm_k` attack in Al-Dujaili et al. [2018] and call an adversarial attack successful if any of the attack algorithms evades the detection.

***Model and Baselines.*** We compare four ML-based malware detectors. The **Unified** detector uses our unified framework in Sec 4.3: we normalize over equivalent API groups, and then adversarially trains over API addition. The **Adv-Trained** detector is adversarially trained with the best adversarial example generated using GREEDYBYGRAD and `rfgsm_k` additive attack. We also include the model proposed by Al-Dujaili et al. [2018], which is adversarially trained against only `rfgsm_k` additive attack, and lastly a **Natural** model with no defense. We use the same network architecture as Al-Dujaili et al. [2018], a fully-connected neural net with three hidden layers, each with 300 ReLU nodes, to set up a fair comparison. We train each model to minimize the negative log-likelihood loss for 20 epochs, and pick the version with the lowest validation loss. We run five different data splits.

***Results.*** As Table 2 shows, relational attacks are overwhelmingly effective to detectors that are oblivious to potential transformations. Adversarial examples almost always (>99% FNR) evade the naturally trained model, and also evade the detector in Al-Dujaili et al. [2018] most of the time (>89% FNR) as it does not consider API substitution. On the defense end, **Unified** achieves the highest robust accuracy: the evasion rate (FNR) only increases by $0.5\%$ on average. **Adv-Trained** comes second

but the evasion rate is $> 20\%$ higher. The evasion is mostly caused by GREEDYBYGROUP, the attack that is too computationally expensive to be included in the training loop. This result corroborates with the theoretical advantage of *N&P*: its robustness guarantee is independent of training algorithms. Last, all detectors using robust learning techniques have higher FPR compared to **Natural**, which is expected because of the inevitable robustness-accuracy trade-off. However, the difference is much smaller compared to the cost due to attacks, and thus the trade-off is worthwhile.

## 6  Conclusion and Future Work

In this work, we set the first step towards robust learning against relational adversaries: we theoretically characterize the conditions for robustness and the sources of robustness-accuracy trade-off, and propose a provably robust learning framework. Our empirical evaluation shows that input normalization can significantly enhance model robustness. For future work, we see automatic detection of semantics-preserving transformation as a promising addition to our current expert knowledge approach, and plan to extend the normalization approach to model explanability, fairness and security problems beyond relational adversaries.

## 7  Societal Impact

We propose input normalization as a principle approach to enhance ML model robustness against relational adversaries. In most cases, the extra robustness in security-critical tasks will reduce the loss caused by malicious behaviors and thus bring positive societal impact. We do acknowledge that one of our specific empirical evaluation setting — automated authorship attribution — can cause privacy issues when used for censorship; an enhanced authorship attribution technique may allow the censoring agent to better identify the author. We note that *N&P* in our experiment only normalizes over the most basic set of program features to investigate ML models' stability in common use cases. For more privacy-sensitive cases, an author can still use more advanced techniques, such as randomize variable/function names, encrypt its code or even normalize its code with a more comprehensive relation *before* code submission, to remove the idiosyncrasies in its codes. Our *N&P* framework over generic program features will not conflict with these anonymization techniques.

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
