# A  Proofs and Explanation for Theoretical Results

In this section, we present the omitted proofs for theorems and observations due to page limit of the main body.

## A.1  Proof for Computational Hardness of Adversarial Training

We first write the full statement of hardness in the following theorem.

**Theorem 2.** *Let $\mathcal{R} \subseteq \{0,1\}^d \times \{0,1\}^d$ be a relation. Given a function $f$, an input $\mathbf{x} \in \{0,1\}^d$ and a feasible set $\mathcal{T}(\mathbf{x}) = \{\mathbf{z} : \mathbf{x} \rightarrow_{\mathcal{R}}^* \mathbf{z}\}$, solving the following maximization problem:*

$$\max_{\mathbf{z} \in \mathcal{T}(\mathbf{x})} l(f, \mathbf{z}, y)$$

*is PSPACE-hard when $l(f, \mathbf{x}, y)$ is the 0-1 classification loss.*

*Proof.* Let $\alpha : \{0,1\}^d \rightarrow \{0,1\}$ be a predicate. Define a loss function $l(f, \mathbf{z}, y)$ as follows: $l(f, \mathbf{z}, y) = \alpha(\mathbf{z})$ (the loss function is essentially the value of the predicate). Note that $\max_{\mathbf{z} \in \mathcal{T}(\mathbf{x})} l(f, \mathbf{z}, y)$ is equal to 1 iff there exists a $\mathbf{z} \in \mathcal{T}(\mathbf{x})$ such that $\alpha(\mathbf{z}) = 1$. This is a well known problem in model checking called *reachability analysis*, which is well known to be PSPACE-complete (the reduction is from the problem of checking emptiness for a set of DFAs, which is known to be PSPACE-complete Kozen [1977]). $\qquad\square$

Recall that the maximation problem $\max_{\mathbf{z} \in B_p(\mathbf{x}, \epsilon)} l(f, \mathbf{z}, y)$ used in adversarial training for the image modality was proven to be NP-hard Katz et al. [2017]. Hence it seems that the robust optimization problem in our context is in a higher complexity class than in the image domain.

## A.2  Model Capacity Requirement

In this section, we illustrate how the *N&P* framework can potentially help reduce the model complexity for learning a robustly accurate classifier. We start with the following proposition.

**Proposition 2** (Model Capacity Requirement). *For some hypothesis class $\mathcal{H}$ and relation $\mathcal{R}$, there exists $f \in \mathcal{H}$ such that $f(\mathcal{N}_{\mathcal{R}}(\cdot))$ is robustly accurate, but no $f \in \mathcal{H}$ can be robustly accurate on the original inputs. In other words, robustly accurate classifier can only be obtained after normalization.*

We first define an equivalence relation induced by equivalent coordinates over binary inputs, and then write the formal statement of the observation in the following claim.

**Definition 5** (Equivalence relation induced by equivalent coordinates). *Let $\mathbf{x} = (\mathbf{x}_1, \cdots, \mathbf{x}_d)$ be a binary input vector on $\{0,1\}^d$, where each $\mathbf{x}_i, i \in \{1, \cdots, d\}$ is a coordinate. Let $I = \{1, \cdots, d\}$ be the set of coordinate indices for inputs in $\mathcal{X}$ and $U = \{i_1, \cdots, i_m\} \subseteq I$. In an equivalence relation $\mathcal{R}$ induced by $U$, $\mathbf{x} \rightarrow_{\mathcal{R}} \mathbf{z}$ iff 1) $\mathbf{x}_i = \mathbf{z}_i$ for all $i \in I \backslash U$, and 2) $\bigvee_{i \in U} \mathbf{x}_i = \bigvee_{i \in U} \mathbf{z}_i$. Notice that $\mathbf{x} \rightarrow_{\mathcal{R}} \mathbf{z}$ iff $\mathbf{z} \rightarrow_{\mathcal{R}} \mathbf{x}$.*

The notation $\bigvee_{i \in U} \mathbf{x}_i$ means taking a *logic or* operation over all $\mathbf{x}_i$s for $i \in U$. Intuitively, having an equivalence relation induced by coordinates with indices in $U$ means the presence of any combination of such coordinates is equivalent to any other combination. Usage of interchangeable APIs in malware implementation is an example of equivalence relation: the attacker can choose any combination from a set of equivalent APIs to implement the same functionality.

In Definition 5, we use $U$ to represent the set of *indices* of the equivalent coordinates. In the following theorems and proofs, we overload $U$ to also represent the set of equivalent coordinates directly, and the $\bigvee$ operation will be taken over all coordinates in $U$.

**Claim 1.** *Consider $\mathcal{X} = \{0,1\}^5$ and $\mathcal{Y} = \{0,1\}$. Let the coordinates of an input $\mathbf{x} \in \mathcal{X}$ be $\{\mathbf{x}_1, \mathbf{x}_1', \mathbf{x}_2, \mathbf{x}_3, \mathbf{x}_4\}$. Suppose we have an equivalence relation induced by $U = \{\mathbf{x}_1, \mathbf{x}_1'\}$. Meanwhile, the true label $y$ of an input $\mathbf{x}$ is 1 iff any of the following clauses is true: 1) $(\mathbf{x}_2 = 1) \wedge (\mathbf{x}_3 = 1)$, 2) $(\mathbf{x}_1 \vee \mathbf{x}_1' = 1) \wedge (\mathbf{x}_2 = 1)$, 3) $(\mathbf{x}_1 \vee \mathbf{x}_1' = 1) \wedge (\mathbf{x}_3 = 1) \wedge (\mathbf{x}_4 = 1)$. Then*

    *1. no linear model can classify the inputs with perfect robust accuracy, but*

    *2. a robust and accurate linear model exists under normalize-and-predict.*

*Proof.* Let $\mathcal{H} = \{f_{\mathbf{w},b} : sgn(\langle \mathbf{w}, \mathbf{x} \rangle + b)\}$. Let $\mathbf{w}_1, \mathbf{w}'_1, \mathbf{w}_2, \mathbf{w}_3, \mathbf{w}_4$ denote the coordinates in $\mathbf{w}$ that corresponds to $\mathbf{x}_1, \mathbf{x}'_1, \mathbf{x}_2, \mathbf{x}_3, \mathbf{x}_4$.

We know $y = 1$ if $\mathbf{x}_1 = 1, \mathbf{x}_2 = 1$ and the other coordinates are zero because the second clause in the labeling rule is satisfied. Therefore, in order to classify this input instance correctly, we must have $\mathbf{w}_2 + \mathbf{w}_1 + b > 0$. Since $\mathbf{x}_1$ and $\mathbf{x}'_1$ are equivalent, we should also have $\mathbf{w}_2 + \mathbf{w}'_1 + b > 0$.

Similarly, we know $y = 0$ if $\mathbf{x}_2 = 1, \mathbf{x}_4 = 1$ and the other coordinates are zero because none of the clauses are satisfied. Therefore, we must have $\mathbf{w}_2 + \mathbf{w}_4 + b < 0$.

In order to classify all possible $\mathbf{x}$ correctly, the classifier $f_{\mathbf{w},b}$ must satisfy

$$\mathbf{w}_2 + \mathbf{w}_4 + b < 0 \tag{3}$$
$$\mathbf{w}_2 + \mathbf{w}_1 + b > 0 \tag{4}$$
$$\mathbf{w}_2 + \mathbf{w}'_1 + b > 0 \tag{5}$$
$$\mathbf{w}_3 + \mathbf{w}_1 + \mathbf{w}'_1 + b < 0 \tag{6}$$
$$\mathbf{w}_3 + \mathbf{w}_4 + \mathbf{w}_1 + b > 0 \tag{7}$$

First, by Formula 3, 4 and 5, we have $\mathbf{w}_1 > \mathbf{w}_4$ and $\mathbf{w}'_1 > \mathbf{w}_4$. However, by Formula 6 and 7, we have $\mathbf{w}_1 + \mathbf{w}'_1 < \mathbf{w}_4 + \mathbf{w}_1$, which implies $\mathbf{w}'_1 < \mathbf{w}_4$. Contradition. Therefore, no linear classifier can satisfy all the equations.

On the other hand, if we perform normalization by letting $\mathbf{x}_1 = \mathbf{x}_1 \vee \mathbf{x}'_1$ and removing $\mathbf{x}'_1$, then a classifier $f_{\mathbf{w},b}$ – with $\mathbf{w}_1 = 0.4, \mathbf{w}_2 = 0.7, \mathbf{w}_3 = 0.5, \mathbf{w}_4 = 0.2, b = -1$ – can perfectly classify $\mathbf{x}$. $\qquad \square$

## A.3 Proof for Theorem 1

*Proof.* We prove the statement by contradiction: suppose all classifiers $f \in \mathcal{H}$ with the highest robust accuracy has $f(\mathbf{x}) \neq f(\mathbf{z})$ for some $\mathbf{x}, \mathbf{z}$ in the same equivalence group $\mathcal{E}$, then we must be able to find a $f'$ such that $f'$ is at least as robust accurate as $f$ and $f'(\mathbf{x}) = f'(\mathbf{z})$.

We construct a classifier $f'$ as follows: 1) $f'(\mathbf{x}) = f'(\mathbf{z})$, and 2) $f'(\mathbf{z}') = f(\mathbf{z}')$ for all $\mathbf{z}' \in \mathcal{X}, \mathbf{z}' \neq \mathbf{z}$, i.e. $f'$ and $f$ agree at all inputs except at $\mathbf{z}$. We discuss the robust accuracy of $f$ and $f'$ in the following two complementary cases.

*Case 1.* If $\mathbf{x}$ and $\mathbf{z}$ are not in the adversarial feasible set $\mathcal{T}(\mathbf{x}')$ of an input $\mathbf{x}'$, then $f$ is robust accurate at $\mathbf{x}'$ iff $f'$ is robust accurate at $\mathbf{x}'$. This is obvious as $f(\mathbf{x}'') = f'(\mathbf{x}'')$ for all $\mathbf{x}'' \in \mathcal{T}(\mathbf{x}')$.

*Case 2.* If $\mathbf{x}$ or $\mathbf{z}$ is in $\mathcal{T}(\mathbf{x}')$, then both $\mathbf{x}$ and $\mathbf{z}$ must be in $\mathcal{T}(\mathbf{x}')$. Suppose $\mathbf{x} \in \mathcal{T}(\mathbf{x}')$, then by the definition of relational adversary, we have $\mathbf{x}' \to^*_{\mathcal{R}} \mathbf{x}$ and by the definition of equivalence group, we have $\mathbf{x} \to^*_{\mathcal{R}} \mathbf{z}$. Therefore, we must have $\mathbf{x}' \to^*_{\mathcal{R}} \mathbf{z}$, i.e., $\mathbf{z} \in \mathcal{T}(\mathbf{x}')$. Similarly, $\mathbf{x}$ must be in $\mathcal{T}(\mathbf{x}')$ if $\mathbf{z}$ is in $\mathcal{T}(\mathbf{x}')$. In this case, $f$ cannot be robust accurate at $\mathbf{x}'$: both $\mathbf{x}$ and $\mathbf{z}$ are in $\mathcal{T}(\mathbf{x}')$, and $f(\mathbf{x}) \neq f(\mathbf{z})$; if $f(\mathbf{x}') = f(\mathbf{x})$, then $\mathbf{z}$ will become a successful adversarial example; similarly, if $f(\mathbf{x}') = f(\mathbf{z})$, then $\mathbf{x}$ will be the adversarial example. Since $f$ is already not robust accurate at $\mathbf{x}'$, the robust accuracy of $f'$ cannot be lower than $f$ in this case.

Combining the two cases, we can conclude that $f'$ is at least as robust accurate as $f$ over the underlying data distribution $\mathcal{D}$. The existence of $g$ naturally follows the definition of normalizer by equivalence group: the normalizer $\mathcal{N}$ guarantees the consistency of prediction within each $\mathcal{E}$, and $g$ just need to match $f$'s prediction on all normalized inputs. $\qquad \square$

Figure. 3: Normalizer in action.

# B  Authorship Attribution – Algorithms and Implementation Details

## B.1  Implementation and Infrastructure

We implement source code *normalizer* on top of Clang cla [2016], an open-source C/C++ frontend for the LLVM compiler framework. For fair comparison of running time, we run all experiment series on an Amazon EC2 c5.18xlarge instance with 72 cores and 144GB memory. We train the model using the *sklearn* and *keras* APIs with *tensorflow* backend.

## B.2  Attack Transformations to be Normalized

The evasion and impersonation attack in Quiring et al. [2019] uses a total of 42 transformation options. These transformations can be divided into two categories based on the information needed. The *general* transformations, such as changes in control statement, can be applied to any code without prior knowledge of the target author. In contrast, the *template-based* transformations, which aim to mimic the target author's function/variable naming and type-def habits, will require samples of the target author's code pieces. We considers 35 of the the 42 transformations as listed in Table 3. [7] The selection covers most transformations including API usage, variable declaration, I/O style and control statement. The transformations are also common across attack papers Quiring et al. [2019], Liu et al. [2021]. The only general transformation *not* considered by us is function in-lining, which removes declared functions in the code and move the commands in-line to the caller function. While this may be done for simple online judge submissions, the boundary of usage of a function is in general hard to track in large code projects. In-lining function in one script may raise problems in other scripts that import and call the function. We do not normalize the transformation as it is hardly semantics-preserving Stucki et al. [2020]. The template-based attack transformations *not* considered by our normalizer is function/variable renaming. The function/variable transformer peeks into the codes written by the target of imitation, and then rename the function and variables to match the target author's habit. This attack action requires extensive knowledge to the target author's coding habit. In the extreme case, the attacker may be more familiar to the target author than the learner, and thus makes the evasion inevitable. We do not normalize this transformation as it can significantly obscure the ground truth.

---

[7]We follow the counting method in Quiring et al. [2019] and count by the number of options instead of number of transformers. For example, the input interface transformer has two options – stdin and file; therefore, we count the input interface transformer as having two transformation options.

Table 3: List of code transformations and their corresponding normalized transformation

| Transformer | Family | Transformer Description | Syntactic Feature of Interests (SFoI) | Normalizing Action in $\mathcal{T}_{\mathcal{N}}$ |
|---|---|---|---|---|
| For statement transformer | Control | Replaces a for-statement by an equivalent while-statement | No. of for loops | Transform for loops to while |
| While statement transformer | | Replaces a while-statement by an equivalent for-statement | | |
| If statement transformer | | Split the condition of a single if-statement at logical operands (e.g., && or ||) to create a cascade or a sequence of two if-statements depending on the logical operand | No. of logical predicate in an if-statement | Split all if-statement so that each statement only has one logical predicate. |
| Array transformer | Declaration | Converts a static or dynamically allocated array into a C++ vector object | No. of array that can potentially be converted to C++ vector object | Convert ALL static or dynamically allocated array into a C++ vector object. |
| String transformer | | Array option: Converts a char array (C-style string) into a C++ string object. The transformer adapts all usages in the respective scope, for instance, it replaces all calls to strlen by calling the instance methods size. String option: Converts a C++ string object into a char array (C-style string). The transformer adapts all usages in the respective scope, for instance, it deletes all calls to c str(). | No. of char array | Convert all string to C++ string object. |
| Integral type transformer | | Promotes integral types (char, short, int, long, long long) to the next higher type, e.g., int is replaced by long. | Number of integral type declaration lower than long long | Replace all integral type with long long. |
| Floating-point type transformer | | Converts float to double as next higher type. | No. of float type declaration lower than double | Convert all float to double. |
| Boolean transformer | | Bool option: Converts true or false by an integer representation to exploit the implicit casting. Int option: Converts an integer type into a boolean type if the integer is used as boolean value only | No. of Boolean values | Use int representation in all cases |
| Init-Decl transformer | | Move into option: Moves a declaration for a control statement if defined outside into the control statement. For instance, int i; ...; for(i = 0; i <N; i++) becomes for(int i = 0; i <N; i++). Move out option: Moves the declaration of a control statement's initialization variable out of the control statement. | No. of looping variable declared outside the control statement | Use move-in option for all scenarios. |
| Typedef transformer | | Convert option: Convert a type from source file to a new type via typedef, and adapt all locations where the new type can be used. Delete option: Deletes a type definition (typedef) and replace all usages by the original data type. | No. of user-defined types | Apply the delete option to all typedef. |

| Transformer | Family | Transformer Description | Syntactic Feature of Interests (SFoI) | Normalizing Action in $\mathcal{T}_\mathcal{N}$ |
|---|---|---|---|---|
| Include-typedef transformer | Template | Inserts a type using typedef, and updates all locations where the new type can be used. Defaults are extracted from the 2016 Code Jam Competition. | No. of user-defined types | Apply the delete option to all typedef. |
| Unused code transformer | Declaration | Function option: Removes functions that are never called. Variable option: Removes global variables that are never used. | No. of unused variables & functions | Remove unused variables and functions. |
| Input interface transformer Output | API | Stdin option: Instead of reading the input from a file (e.g. by using the API ifstream or freopen), the input to the program is read from stdin directly (e.g. cin or scanf). File option: Instead of reading the input from stdin, the input is retrieved from a file. | No. of file I/O | Use stdin always. |
| Output interface transformer | | Stdout option: Instead of printing the output to a file (e.g. by ofstream or freopen), the output is written directly to stdout (e.g. cout or printf). File option: Instead of writing the output directly to stdout, the output is written to a file. | | Use stdout always. |
| Input API transformer | | C++-Style option: Substitutes C APIs used for reading input (e.g., scanf) by C++ APIs (e.g., usage of cin). C-Style option: Substitutes C++ APIs used for reading input (e.g., usage of cin) by C APIs (e.g., scanf). | No. of C-style I/O API | Use C++ API always |
| Output API transformer | | C++-Style option: Substitutes C APIs used for writing output (e.g., printf) by C++ APIs (e.g., usage of cout). C-Style option: Substitutes C++ APIs used for writing output (e.g., usage cout) by C APIs (e.g., printf). | | Use C++ Style always. |
| Sync-with-stdio transformer | | Enable or remove the synchronization of C++ streams and C streams if possible | No. of potential synchronization sites | Enable all synchronization. |
| Compound statement transformer | Misc | Insert option: Adds a compound statement ({...}). The transformer adds a new compound statement to a control statement (if, while, etc.) given their body is not already wrapped in a compound statement. Delete option: Deletes a compound statement ({...}). The transformer deletes compound statements that have no effect, i.e., compound statements containing only a single statement. | No. of compound statements | Apply the delete option to all locations. |
| Return statement transformer | | Adds a return statement. The transformer adds a return statement to the main function to explicitly return 0 (meaning success). Note that main is a non-void function and is required to return an exit code. If the execution reaches the end of main without encountering a return statement, zero is returned implicitly. | No. of implicit return sites | Adds the return statement to all applicable locations. |

## B.3 Order Function $\lambda$ & Normalizing Transformer $\mathcal{T}_\mathcal{N}$

We identify the Syntactic Features of Interests (SFoI) corresponding to the transformers as shown in Table 3. Each SFoI's value can be changed by one or multiple transformers. For example, we use the number of for loops in the code as one SFoI, and its value can be changed by two transformers – the *for-to-while* and the *while-to-for* transformer over the control flow.

We take the order function $\lambda$ as the *sum* of values of the SFoIs, and the normal form of a code piece is the variant that has the smallest sum of SFoIs. For example, if the SFoIs are 1) number of *for* loops and 2) number of *printf* statements, then the normal form will be the variant that has the least number of *for* loops and *printf* statements in total. In Table 3, we identify a total of 16 SFoIs and use the sum of all of them as our order function $\lambda$.

We select a subset of the attack transformations as the set of normalizing transformations $\mathcal{T}_\mathcal{N}$ so as to respect the equivalence groups. Our $\mathcal{T}_\mathcal{N}$ contains and only contains the attack transformations that strictly decrease the value of SFoIs. For example, when the number of *for* loops is used as an SFoI, we keep the *for-to-while* transformer in $\mathcal{T}_\mathcal{N}$ and discard the *while-to-for*. The right-most column of Table 3 shows all the transformations we keep in $\mathcal{T}_\mathcal{N}$. Notice that the SFoIs all take non-negative integer values, and the normalizing transformations reduce the value of $\lambda$ by 1 once applied. Therefore, the number of iterations in normalization for an input $\mathbf{x}$ is bounded by the value of $\lambda(\mathbf{x})$.

## B.4 Normalization in Action

Figure 3 shows an example of our normalizer in action. The left-most box contains an code snippet originally written by Author A. The subsequent code boxes in the top row illustrate a sequence of transformations applied to the original code. The attacker first converts the C-style *printf* statement to the C++ style *cout* statement, then changes the *for* loop to a *while* loop, and eventually changes the value of a boolean variable from 0 to False. The final variant in right-most box is a successful adversarial example that misled the model to predict a different author.

In this code example, three syntactical features of interests are involved: 1) the number of *for* loops, 2) the number of C-style I/O statements and 3) the number of Boolean values that can be cast into integers. The normalizer applies the normalizing actions in a iterative manner, reducing the number of *for* loops, C-style IO statements and Boolean values until no more action is applicable. All four code pieces – the original input, the final adversarial example and the two intermediate variants – will be normalized into the same normal form as depicted in the bottom box in Figure 3.

## B.5 Adaptation for Adversarial Training

In existing adversarial training literature, the $\ell_p$-norm based adversarial examples are created directly in the feature-space using gradient ascent; the attack can be readily computed in GPU in a similar manner as model updates. The code transformations, however, are performed in the problem-space; the MCTS computation is CPU-intensive and thus takes much longer. In addition, the validity check of adversarial examples further increases the computational load. We make the following adjustments to speed up the adversarial training process. First, we use **Vanilla-LSTM** as the initial model and fine-tune it using adversarial inputs. The models show improved robust accuracy (~60%) to the attacks in the training loops after 10 epochs. Second, instead of generating adversarial examples at every training step, we generate the adversarial training inputs for all .cpp files with respect to the model at beginning of an epoch. This change allows us to generate the adversarial training inputs in parallel. To further speed up the adversarial training procedure, we also reduce the max-depth from 25 to 5 as well as the number of random play-outs at each node from 50 to 10 in the Monte-Carlo tree-search. With these modification, we finally manage to finish adversarial training within a month.

# C  Malware Detection – Algorithms and Implementation Details

In this section, we present the omitted algorithm descriptions and experiment implementation details for the malware detection experiment.

## C.1  Generic Relational Attack Algorithms

In Sec 5, we introduce two generic relational attack algorithms – GREEDYBYGROUP and GREEDYBYGRAD. The algorithm boxes below shows the exact description of both algorithms.

---

**Algorithm 1** GREEDYBYGROUP $(\mathbf{x}, y, K)$

---

$\mathbf{x}^{adv} = \mathbf{x}, k = 0$
Partition $\mathcal{R}$ into $m$ groups $\{\mathcal{R}_1, \cdots, \mathcal{R}_m\}$.
**while** $k < K$ **do**
    $k = k + 1$
    **for** $\mathcal{R}_i \in \{\mathcal{R}_1, \cdots, \mathcal{R}_m\}$ **do**
        $\mathbf{x}_i = \arg \max\limits_{\mathbf{z}: \mathbf{x}^{adv} \to^*_{R_i} \mathbf{z}} \ell(f, \mathbf{z}, y).$
    **end**
    Combine $\mathbf{x}_i$s to obtain the new $\mathbf{x}^{adv}$
**end**
**return** $\mathbf{x}^{adv}$

---

**Algorithm 2** GREEDYBYGRAD$(\mathbf{x}, y, m, K)$

---

$\mathbf{x}^{adv} = \mathbf{x}, k = 0$
**while** $k < K$ **do**
    $k = k + 1$
    $g = \nabla_{\mathbf{x}} \ell(f, \mathbf{x}^{adv}, y)$
    **for** $(\mathbf{x}^{adv}, \mathbf{z}) \in \mathcal{R}$ **do**
        $c_{(\mathbf{x}^{adv}, \mathbf{z})} = \sum_{i=1}^{d} g_i (\mathbf{z}_i - \mathbf{x}_i^{adv})$
    **end**
    Apply the transformations with top $m$ largest positive $c_{(\mathbf{x}^{adv}, \mathbf{z})}$ to obtain the new $\mathbf{x}^{adv}$.
**end**
**return** $\mathbf{x}^{adv}$

---

## C.2  GREEDYBYGRAD and GREEDYBYGROUP for Malware Detection on Sleipnir

We instantiate the two attacks on our malware detection task as follows. For GREEDYBYGROUP, we divide the relation by equivalent API groups. In each iteration, the attacker searches the best combination of API in each equivalence group that causes the most increase in test loss. These combinations are concatenated using a logical **OR** operation. For GREEDYBYGRAD, the attacker tries all single API substitutions and additions allowed by the relation, and use the first-order approximation of test loss to determine the $m$ transformations to be applied in each iteration. We run both attacks for various $K$ and $m$. We find that $K = 10$ and $m = 10$ suffices to bypass detection for the vanilla model.

## C.3  Extract API Substitution Rules

In Sec 5, we consider malware authors who can substitute API calls with equivalent API calls to evade ML-based malware detector. We now explain how we extract the equivalent APIs. We identify four types of patterns for extracting equivalent APIs:

- API with the same name but located in different Dynamically Linkable Libraries (DLLs). For example, `memcpy`, a standard C library function, is shipped in libraries with different names, including `crtdll.dll`, `msvcr90.dll`, and `msvcr110.dll`.
- API with and without the `Ex` suffix. The `Ex` suffix represents an extension to the same API without the suffix.
- API with and without the `A` or `W` suffixes. The `A` suffix represents the single character version. The `W` suffix represents the wide character version.
- API with/without `_s` suffix. The `_s` suffix represents the secure version of an API.

Using these four patterns, we extracted about 2,000 equivalent API groups. About 500 of the groups have more than 2 APIs and the maximal group has 23 APIs.