# OpenReview forum: "Robust Learning against Relational Adversaries"
_NeurIPS.cc/2022/Conference — NeurIPS 2022 Accept_

### Official Review · Reviewer_7d2p · 2022-07-06

**Rating:** 8
**Confidence:** 3
**Soundness:** 4 excellent
**Presentation:** 4 excellent
**Contribution:** 4 excellent

**Summary:**

The paper studies robustness against relational adversaries, a broad class of attackers who create adversarial examples in a reflexive-transitive closure of a logical relation and is motivated by attacks in program analysis and security tasks. Specifically, the paper first analyzes the conditions for robustness against relational adversaries and investigates different levels of robustness-accuracy trade-off.  Inspired by the insights, the authors propose normalize-and-predict (N & P), a learning framework that leverages input normalization to achieve provable robustness by design. The framework can be also combined with adversarial training ti further enhance the robustness. The proposed framework is tested on source code authorship attribution and malware detection and significantly improves the robustness of models against relational adversaries.

**Questions:**

How can you generate the “equivalence group” in general real-world problems? Is there any possible automatic way?

How accurate is the proposed efficient normalization, compared to the exact normalization?

Can you comment on: whether and how the robust learning results of the N & P + AdvTrain can be used to provide model explanations?

**Limitations:**

The authors adequately addressed the limitations and potential negative societal impact.

**Strengths And Weaknesses:**

Strengths

+ Paper is well-written and well-organized

+ The studied problem is novel

+ The proposed N & P is novel and theoretically grounded



Weaknesses

- I do not see any serious weaknesses

---

> ### Author Response · Authors · 2022-08-01
> **Author Response**
>
> We thank the reviewer for the insightful questions, and will answer these questions to the best of our knowledge in this response.
>
> > *How can you generate the “equivalence group” in general real-world problems? Is there any possible automatic way?*
>
> In Section 4.2, an equivalence group is defined as the maximal set containing semantically equivalent inputs under relation $\mathcal{R}$. It is a useful concept for describing the ideal normalizer and analyzing its performance. In real-world problems, however, we do not explicitly generate an equivalence group because it could have a size exponential to the input dimension. Instead, we assign an order to elements in an equivalence group and attempt to find the element with the lowest order as the normal form. Nonetheless, equivalence groups still implicitly affect our construction of the normalizer. As a boundary condition, the normalizer should never transform an input outside its equivalence group.
>
> On the other hand, we feel that automatically generating the relation itself could be interesting. In our empirical evaluation, we use expert knowledge to generate transformation rules. Is there a data-driven way of finding transformation rules? Meanwhile, the transformations can also be parametrized on the latent feature space for faster computation. These are interesting open questions to explore in the future.
>
> > *How accurate is the proposed efficient normalization, compared to the exact normalization?*
>
> First, we note that for malware detection, normalization over equivalent API groups is exact. For authorship attribution, we measure the effectiveness of our heuristic normalizer by $||x_{adv} - x||/||x||$, the norm ratio of the perturbation over the original input on the feature space. Our heuristic normalizer can reduce this ratio by more than 70% on average. The normalized adversarial example is much closer to the normalized clean input compared to no normalization.
>
> > *Can you comment on: whether and how the robust learning results of the *N\&P* + AdvTrain can be used to provide model explanations?*
>
> First, we feel that the relational attack itself could be a method to generate counterfactual explanation. By defining a set of explainable transformations on the input space, the sequence of transformations for a successful adversarial example naturally shows how the input can be modified to obtain a different prediction. For example, for loan applications, this method can inform the applicant of the areas of improvements to obtain a larger loan.
>
> Second, *N\&P* could potentially be useful for AI fairness. Ensuring the model is not discriminating by factors such as race and gender is similar to enforcing invariance of prediction over these factors. The challenge is, however, that the relation may not only involve the apparent factors in the inputs. For example, removing race/ethnicity information alone may not be sufficient when features like post code are highly correlated with sensitive demographic information. In this case, finding and normalizing relations on the latent feature space may help. (Compared to DANN and other end-to-end methods, this approach may be more “generative” as it assumes a relational structure on the latent space.)
>
> Thank you for reading the response. Model explanation is not our primary research area, so the ideas might be somewhat naïve. Nevertheless, we hope the discussion is helpful!

---

> > ### Comment · Reviewer_7d2p · 2022-08-05
> > **Reply to authors' rebuttal**
> >
> > Thanks a lot for the response. It is more clear to me now.

---

> > > ### Author Response · Authors · 2022-08-06
> > > **Thank you for your prompt response**
> > >
> > > We appreciate your feedback in both the review and the discussion. Thank you!

---

### Official Review · Reviewer_fYgj · 2022-07-11

**Rating:** 9
**Confidence:** 4
**Soundness:** 4 excellent
**Presentation:** 3 good
**Contribution:** 4 excellent

**Summary:**

The authors first provide the definition of relational adversaries and then propose normalize-and-predict framework for a robust classifier against those adversaries. They show how to obtain the optimal normalizer for normalizing the model's input. Then, they apply the adversarial training to deal with the robustness-accuracy tradeoff. As a result, their approach can outperform state-of-the-art defenses (e.g., adversarial training).

**Questions:**

I have no question regarding this paper.

**Limitations:**

The authors provide how they deal with the negative societal impact very well.

**Strengths And Weaknesses:**

Strength
The approach is promising with the theoretical proof for the normalizer. Also, it can empirically outperform the state-of-the-art defenses in terms of robustness and the tradeoff.

Weakness
There is only two datasets in the experiment. It would be better to see more datasets.

---

> ### Author Response · Authors · 2022-08-01
> **Author Response**
>
> We thank the reviewer for the encouraging comments! As part of our ongoing project, we are 1) further improving the computation efficiency of our pipeline so as to use it for datasets at order-of-magnitude larger scale, and 2) investigating normalization at different levels of representation. We look forward to sharing more exciting results with the community in the future.

---

### Official Review · Reviewer_2Ca5 · 2022-07-18

**Rating:** 6
**Confidence:** 2
**Soundness:** 4 excellent
**Presentation:** 3 good
**Contribution:** 3 good

**Summary:**

This paper investigates relational adversaries, which craft adversarial examples via an arbitrary number of transformations. Based on the theoretical analysis of the robustness-accuracy trade-off, the authors propose normalize-and-predict, a learning framework that leverages input normalization to achieve provable robustness. Experiments on source code authorship attribution and malware detection verify the proposed method.

**Questions:**

Is the reason why there is a trade-off between robustness and accuracy in defending relational opponents different from $\ell_p$ attacks?

**Limitations:**

A major limitation is that the effectiveness of the proposed framework depends heavily on specified transformations, some of which may be unknown to the defender.

**Strengths And Weaknesses:**

### Strengths
- The considered adversarial robustness against relational adversaries is well defined. This problem would be important to the community.
- The three levels of the robustness-accuracy trade-off are interesting and insightful. The second and third cases indeed happen in practice.

### Weaknesses
- Although it is well-known that there might exist a robustness-accuracy trade-off in defending against $\ell_p$ attacks, it is still necessary to cite relevant papers in the main text for reference. Moreover, while it is true that robustness against $\ell_p$ attacks may be at odds with accuracy, it does not mean that the trade-off is inevitable. For example, human visual systems are known to be simultaneously robust and accurate under small $\ell_p$ perturbations.
- This paper claims that the robustness-accuracy trade-off exists for learning against relational attacks and depicts three levels. However, the claimed trade-off relies on relational adversaries being able to change labels with feasible transformations. This is very different from the $\ell_p$ attacks in the image domain, where small perturbations hardly change the ground-truth labels. If the ground-truth labels are allowed to be different, then is not surprising that the trade-off occurs.
- The effectiveness of the proposed framework depends heavily on the specified transformations. It would be unrealistic to assume that all the transformations the attacker can apply can be known in advance.

== Post rebuttal ==
Based on the rebuttal, I have decided to increase my score to weak accept.

---

> ### Author Response · Authors · 2022-08-01
> **Author Response, Part 2**
>
> > *The effectiveness of the proposed framework depends heavily on the specified transformations. It would be unrealistic to assume that all the transformations the attacker can apply can be known in advance.*
>
> We admit that in security tasks, the attacker can have access to more resources and knowledge than the defender. However, in relational attacks, knowing the transformations is necessary to formally analyze the effectiveness of any robust learning pipeline. This is because the set of attack transformations directly determines the feasible set of adversarial examples, like $p$ and $\epsilon$ in $\ell_p$ attacks. Having extra attack transformations in test-time would be analogous to using translation, masking and scaling against an $\ell_p$ defense in image classification. The effectiveness of the defense in such a mismatched scenario can only be checked via empirical evaluation.
>
> We performed such an evaluation for authorship attribution. The **Adaptive+** attack uses more transformations than the learner normalizes. Although not as robust as in the equal knowledge case, *N\&P* still achieves higher robust accuracy than vanilla models, possibly because normalization reduces the attack surface.
>
> In addition, we note that practical real-world attackers may not always be as strong as the worse-case ones. Taking malware as an example. A common type of attackers will conveniently use an existing malware as the base and then apply transformations such as repackaging to evade detection. The transformations are not necessarily the most sophisticated because the attackers 1) have a practical computation budget, 2) lack the full (semantic) knowledge of the base malware to perform more advanced transformations, and/or 3) want to have fast iterations of malware variants to invade more vulnerable targets instead of generate one variant that breaks all detectors. A security expert is likely aware of the common transformations used by an average attacker of this type.
>
> Last, although dubbed as relational “adversaries”, the threat model also has implications in non-adversarial settings such as AI fairness. From a learner’s perspective, the set of transformations represents the invariance properties expected on model predictions. For example, a fraud detection model should not discriminate based on race or gender. A learner can use relations to specify these factors. In this case, the transformations are defined to match the "natural adversary" — the possible data variation over the sensitive factors — in test-time.
>
>
> #### **References**
>
> [R1] Zhang, Hongyang, et al. "Theoretically principled trade-off between robustness and accuracy." International conference on machine learning. PMLR, 2019.
>
> [R2] Tsipras, Dimitris, et al. "Robustness May Be at Odds with Accuracy." International Conference on Learning Representations. No. 2019. 2019.
>
> [R3] Fawzi, Alhussein, Omar Fawzi, and Pascal Frossard. "Analysis of classifiers’ robustness to adversarial perturbations." Machine learning 107.3 (2018): 481-508.
>
> [R4] Yang, Yao-Yuan, et al. "A closer look at accuracy vs. robustness." Advances in neural information processing systems 33 (2020): 8588-8601.
>
> [R5] Raghunathan, Aditi, et al. "Understanding and mitigating the tradeoff between robustness and accuracy." Proceedings of the 37th International Conference on Machine Learning. 2020.
>
> [R6] Quiring, Erwin, Alwin Maier, and Konrad Rieck. "Misleading authorship attribution of source code using adversarial learning." 28th USENIX Security Symposium (USENIX Security 19). 2019.

---

> > ### Comment · Reviewer_2Ca5 · 2022-08-05
> > **Thanks**
> >
> > I thank the authors for the detailed response. My concerns have been addressed, especially the trade-off between robustness and accuracy. The discussion on the literature and human visual systems is insightful. Thus, I would like to increase my score from 5 to 6.

---

> > > ### Author Response · Authors · 2022-08-06
> > > **Thank you**
> > >
> > > We appreciate your prompt response! Thank you once again for your comments and we will add the related work after the reversion.

---

> ### Author Response · Authors · 2022-08-01
> **Author Response, Part 1**
>
> We thank the reviewer for the insightful comments! The reviewer questions 1) whether and why the robustness-accuracy trade-off against relational adversaries is different from $\ell_p$ attacks, and 2) the extent to which the *N\&P* framework depends on knowing the attack transformations. We organize our discussion around the three weaknesses identified in the review. The first two responses discuss the trade-off; the last discusses the learner’s knowledge on transformations.
>
> > *Although it is well-known that there might exist a robustness-accuracy trade-off in defending against $\ell_p$ attacks, it is still necessary to cite relevant papers in the main text for reference. Moreover, while it is true that robustness against $\ell_p$ attacks may be at odds with accuracy, it does not mean that the trade-off is inevitable. For example, human visual systems are known to be simultaneously robust and accurate under small $\ell_p$ perturbations.*
>
> We will definitely discuss related robustness-accuracy trade-off work in the revised version. Thank you for pointing out! In case the following discussion misses any important reference, please kindly let us know.
>
> The robustness-accuracy trade-off against $\ell_p$ attacks has been discussed in literature including [R1, R2, R3, R4, R5]. The trade-off can either be 1) distributional, where inputs with different Bayes-optimal labels are mixed up in the adversarial feasible set, or 2) algorithmic, where the trade-off is due to low model expressiveness, small sample size, inductive bias in training data or deficiency in the learning algorithm. The former is inevitable because no classifier can achieve both optimal robustness and accuracy. Zhang et al. construct such an example distribution in [R1]: the real number line is divided into intervals of size $\epsilon$, and neighboring segments always have different labels.
>
> Our analysis focuses on distributional trade-off because *N\&P* guarantees robustness against the normalized transformations, i.e. there is no further algorithmic trade-off if normalization is exact. The analysis helps strategically choose the set of transformations to normalize in the unified framework. We also note that *N\&P* can use any existing training/testing procedure except the inputs are now normalized. Therefore, existing $\ell_p$-based trade-off analysis also applies on the normalized input space for $\ell_p$-attacks.
>
> The human visual systems are known to be both robust and accurate in most cases. In fact, [R4] has shown that CIFAR and Restricted ImageNet datasets are well separated between classes in $\ell_p$ distance. However, there are also hard cases for human eyes. E.g., a cat and a dog are easy to distinguish from the front view but not as easy from the back. (I was fooled by this [image](https://www.shutterstock.com/image-photo/rear-view-shetland-sheepdog-sitting-front-346701881).) Distributional trade-off, although less common, can still happen to humans.
>
> > *This paper claims that the robustness-accuracy trade-off exists for learning against relational attacks and depicts three levels. However, the claimed trade-off relies on relational adversaries being able to change labels with feasible transformations. This is very different from the $\ell_p$ attacks in the image domain, where small perturbations hardly change the ground-truth labels. If the ground-truth labels are allowed to be different, then it is not surprising that the trade-off occurs.*
>
> First, we note that the transformations do not change the label of a specific training/testing instance. E.g., a malware remains malicious after transformations. To be precise, the input is transformed to a form $x$ that is more likely to be benign on the distribution, i.e. $\Pr(Y=benign|x) > \Pr(Y=malicious|x)$.
>
> Such cases happen more often in program analysis against relational adversaries than in image domain against $\ell_p$ attacks for two reasons. First, relational attacks preserve the essential semantics rather than the syntax. The perturbations can have much larger $\ell_p$ norm than in image tasks [R6]. Second and more fundamentally, as a compromise for computation overhead, the feature representation in static program analysis may not fully capture the runtime behavior of the program. For example, ransomware may use the same APIs and libraries as zip tools. To distinguish the two, one has to collect their runtime traces or perform symbolic execution, both of which are too computationally demanding for a lightweight virus scan tool.
>
> In fact, the second reason is a key motivation of our work — we want to understand how much false sense of security we are getting in existing learning pipelines. By normalizing the input w.r.t. common input transformations, we can eliminate the accuracy gain from spurious correlation with mere syntactical features. The robust accuracy more faithfully reflects the model’s effectiveness for security-critical tasks in real adversarial environments.

---

### Meta-Review · Area_Chair_31sk · 2022-08-22

**Recommendation:** Accept
**Confidence:** Certain

**Metareview:**

This paper studies relational adversary, a general threat model in which adversary can manipulate test data via transformations specified by a logical relation. Inspired by the conditions for robustness against relational adversaries and the sources of robustness-accuracy trade-off, the authors propose a learning framework called normalize-and-predict, which leverages input normalization to achieve provable robustness. Experiments on two tasks verify the effectiveness of the proposed method.

Overall, the paper is well organized and presented. It proposes a novel and technically sound defense mechanism and takes the first step towards robust learning against relational adversaries. The author responses have address reviewers' concerns, and all reviewers finally agree on the acceptance.

**Award:**

No

---

### Decision · Program_Chairs · 2022-09-14

Accept